# Analysis of the Spatial Variations of Determinants of Gully Agricultural Production Transformation in the Chinese Loess Plateau and Its Policy Implications

**Lulu Qu [1], Yurui Li [2],\*, Yunxin Huang [2], Xuanchang Zhang [2] and Jilai Liu [3]**

[1] School of Public Affairs, Chongqing University, Chongqing 400044, China; qululu91@cqu.edu.cn
[2] Institute of Geographic Sciences and Natural Resources Research, Chinese Academy of Sciences, Beijing 100101, China; huangyx.20b@igsnrr.ac.cn (Y.H.); zhangxc.18b@igsnrr.ac.cn (X.Z.)
[3] School of Geographic Sciences, Xinyang Normal University, Xinyang 464000, China; liujl@xynu.edu.cn
\* Correspondence: liyr@igsnrr.ac.cn

**Abstract:** Exploring the gully agricultural production transformation and its influencing factors is of considerable significance to the evolution of the human–land relationship and multifunctional transformation of gully agriculture in the context of new development. This paper tries to reveal intensive land use under the background of population contraction in the Chinese Loess Plateau and its transformation trend by defining the gully agricultural production transformation (GAPT). Given the representativeness of land-use change in the loess hilly and gully region (LHGR) was taken as a case study, and ArcGIS spatial analysis techniques and geographically and temporally weighted regression model (GTWR) were used to detect the spatio-temporal differentiation pattern and influencing factors. The results show that: (1) GAPT shifts from the high elevation area of 1000–1300 m to the low elevation area of <1000 m, and the transformation process remains within the range of slope 0–20° and topographic relief between 40 m and 180 m. (2) GTWR coupled with time non-stationary and spatial heterogeneity has a better fitting effect, which verifies its applicability in the study of GAPT. Social and economic factors were the main driving forces of GAPT in Yan'an City in the past 20 years, and they were increasing year by year. (3) The spatial-temporal distribution of the driving factors of the agricultural production transformation in Yan'an City is different. The intensity of the population factor and the slope factor is always in the dominant position, and the high value distribution area of the land average GDP factor forms a funnel-shaped pattern of "core edge" in the north and the central and western regions, and its changes tend to "flow" to the core. (4) The gully agricultural production transformation can reflect the general law of rural land use transition in gully areas, and thereby provide policy ideas for gully development. Overall, this study's content can provide scientific guidance for the sustainable development of gully agriculture and the revitalization of watershed and land consolidation in gully areas.

**Keywords:** gully agricultural production transformation; rural development; sustainable land use; geographically and temporally weighted regression; gully land consolidation



## 1. Introduction

The Loess Plateau is a special region integrating agriculture from farming areas to pastoral areas, ecologically fragile areas, and economically poor areas in China. Over a long period of time, the irrational use of local resources has caused vegetation degradation, soil erosion, and severe land productivity reduction [1,2], reflecting the contradiction between ecological protection and economic development, and a key area connecting precise poverty alleviation and Rural Revitalization [3]. Ecological protection, human land system coordination, and sustainable development are always the basic propositions of high-quality development in the Loess Plateau [4,5]. The loess hilly and gully region (LHGR) is characterized by undulating hills and gullies, and its unique geographical

characteristics have shaped a distinctive rural man land system [6]. In the process of social and economic development and urbanization, the loess hilly and gully region has suffered from the dual disturbance of natural ecology and human activities. With the implementation of the Grain for Green Project (GGP) and Gully Land Consolidation (GLC), the regional vegetation has been significantly improved, and the contradiction between people, food, and land has been gradually eased, the transformation and development trend of rural man land system in the loess hilly and gully region is obvious [7].

Since the reform and opening up, the operation characteristics of the rural man land system in the LHGR can be summarized into three stages: sloping agriculture stage, vegetation construction stage, and gully agriculture development stage. The process has involved the extensive planting and low income of agricultural production and then the sustainable saving of production practice, and the overall trend of transformation to modern agriculture [8,9]. At present, the related research on the rural man land system in the LHGR mainly focuses on the perspective of new types of business entities, ecological governance and industrialization, focusing on the spatial form of the core elements (settlement and land use) of the rural man land system [5,10,11], typical patterns [12–14], evolutionary process [15,16], dynamic factors and mechanisms [17,18].

Human activity in the Loess Plateau has a long and complicated history of more than a thousand years of human settlement [19]. In the new period, there is a shortage of high-quality cultivated land in the rural areas of the LHGR, the development space of construction land in the valley is limited, the ecological restoration of the gully region coexists with the rural decline, the slow development of the gully countryside and the "rural disease". Five problems, including high-speed non-agricultural transformation, over-fast aging, deep poverty, severe fouling of soil and water environment, and increasingly hollowing land management of the countryside, are more prominent, and the development of the rural human-earth system in the LHGR urgently needs to be reconstructed [20,21]. With the continuous development of Metrology geography and human-earth system science, the research on the evolution of the human-earth system is constantly updated and in-depth, and the mutual feed and correlation between system elements are emphasized. The dynamic analysis of the system and the model methods such as GWR, ESDA, geo-detector, and other model methods has been paid attention to by the academic circles [22,23], gradually embedding spatial factors into the system model to reveal the driving mechanism of the system factors.

The Loess Plateau can be divided into geomorphic zones on the basis of geologic structure and topography. However, terraces have often been built on the slopes of loess Liang or Mao to create fields for agriculture, and check dams have been constructed in gullies and valleys [24]. The evolution of the gully man land system with gully farmland as the core has both temporal and spatial attributes. The changes in temporal and spatial geographical location will cause changes in the relationship or structure among variables [25]. The non-stationarity of time (lag effect) needs to be included in the scope of the model. At the same time, the spatial and temporal dimensions should be included in the driving force analysis model. It is of great significance to explore the spatiotemporal characteristics and laws of the driving forces of rural land conversion. The rural man land system in the LHGR is a system with different spatial changes in nature, economy, and environment. Its focus is more dependent on the agricultural production of the gully farmland unit [26].

In this context, this research focuses on the evolution of the rural man-land system with gully farmland as the core. The key point is to analyze the temporal and spatial relationship between people and land in the system, and integrate it into a framework to explore farmland changes and the spatio-temporal influence mechanism, explore "cure" approaches, and prescribe the right prescriptions to consolidate the effects of the Grain for Green Project and Gully Land Consolidation, thereby contributing to rural revitalization and regional sustainable development.

In view of this, this paper intends to select 30 Landsat TM/OLI data in four periods of 1995–2000, 2000–2005, 2005–2010, 2010–2018, and apply the CART decision tree classifica-

tion algorithm for remote sensing image interpretation to obtain the corresponding time series of gully farmland change information, realizing the identification of gully farmland and the driving force analysis of gully farmland spatio-temporal evolution, analyzing its influence mechanism. Thus, it provides a scientific reference for the sustainable development of gully agriculture and the revitalization of watershed in the Chinese Loess Plateau, especially in the LHGR. Implications for rural development policy related to land consolidation would be also addressed.

## 2. Theoretical Analysis

### 2.1. A Theoretical Model for Gully Agricultural Evolution in Gully Areas

Since the economic reforms and open-door policy were initiated in 1978, the agricultural development in the LHGR has generally experienced three stages: the traditional sloping agriculture stage, transition stage, and modern gully agriculture stage. Firstly, in the traditional sloping agriculture stage, the agricultural production was mainly on the slope, and sloping farmland was the primary source of production in the regional agricultural production system. Due to the relatively high altitude, the agricultural production was large-scale planting and little harvesting, and the land use was extensive. Secondly, with the large-scale GGP implemented in 1999, the sloping land was converted to farmland, and the scale of the area continued to decrease. The slope farming was gradually withdrawn, and vegetation replaced the sloping farmland. The surface transformed from "yellow" to "green", and agricultural production has now shifted from high-altitude slopes to low-altitude gully. The living spaces also gathered from high-altitude areas to low-altitude areas, and the gully had become the key area for the production and living in the LHGR. Thirdly, in order to develop the agricultural production in the gully, the GLCP's "consolidation of the gully to protect the ecology and creating land to benefit the people's livelihood" was implemented, achieving the coordinated development of ecology and production (Figure 1). In addition, through the GLCP, the comprehensive agricultural production capacity was improved, and the integration of "three industries" and the development of agricultural conservation were promoted [21,27].

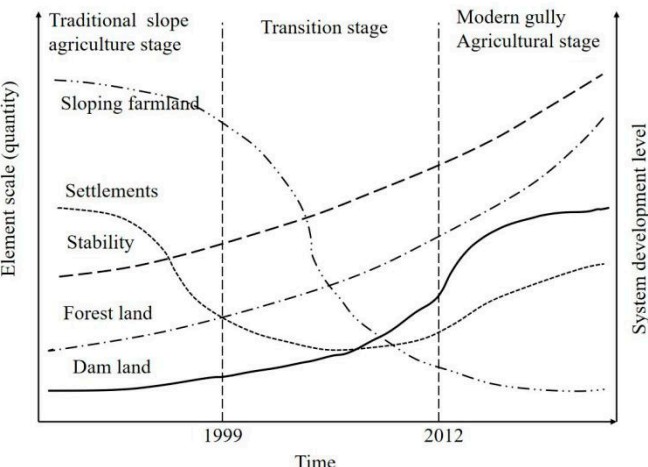

**Figure 1.** The evolution theoretical model of the gully agricultural production development.

### 2.2. The Evolution of the Gully Agricultural in the LHGR

Since 1999, the Grain for Green Project (GGP) in the LHGR, especially the Gully Land Consolidation (GLC) project in 2012, the gully rural human–land relationship has changed subtly), gradually shifting from the traditional sloping agricultural system to new gully agricultural systems; the agricultural system of the LHGR is transforming. The core of this transformation is to achieve the "win-win" goal of ecological and economic benefits, and mainly manifested in two aspects: food crop production of the gully farming system and ecological conservation of the gully forestry system in the LHGR. Thus, changes in land

use patterns correspond to the transformation of functional effects driven by economic and social development and innovation that are compatible with the stage of economic and social development [20,28]. The traditional farming system in the LHGR is centered on the main production of food crops. As time goes by, the gully watershed becomes the distinctive land use pattern of the gully area. Adapting to the stage of social and economic development, agricultural production gradually transforms from extensive development and utilization of land expansion to intensive utilization under an ecological economy. The goal of this study is to deconstruct the multiple elements of the agricultural production system in gully areas from the perspective of farmer subject and regional production space and type change, evaluating the interaction mechanism that drives changes in the human–land relationship in the LHGR.

## 3. Materials and Methods

### 3.1. Geography of the Study Region

The loess hilly and gully region of Yan'an City is located in the center of the Loess Plateau, with a total area of about 18,729 km². It is the combination of the middle and upper reaches of the Yellow River Basin and the northern agro-pastoral region, covering eight districts and counties in the north-central part of Yan'an: Baota District, Yanchang County, Ganquan County, Ansai District, Yanchuan County, Zichang County, Zhidan County, and Wuqi County (Figure 2). The topographical conditions of the study area are complex, with crisscrossing ditches, streams, slopes, beams, and ridges, and the topography is very typical. Since 1999, as the first batch of pilot areas for the Grain for Green project and Gully Land Consolidation, this area has taken the lead in realizing the transformation of the surface color from "yellow" to "green" and the production space from "sloping" to "gully". In the context of urban-rural integration and the high-quality development of the Yellow River Basin, it is typical and representative to carry out the research on the characteristics and mechanism of gully farmland transformation in the loess hilly and gully region of Yan'an City [29].

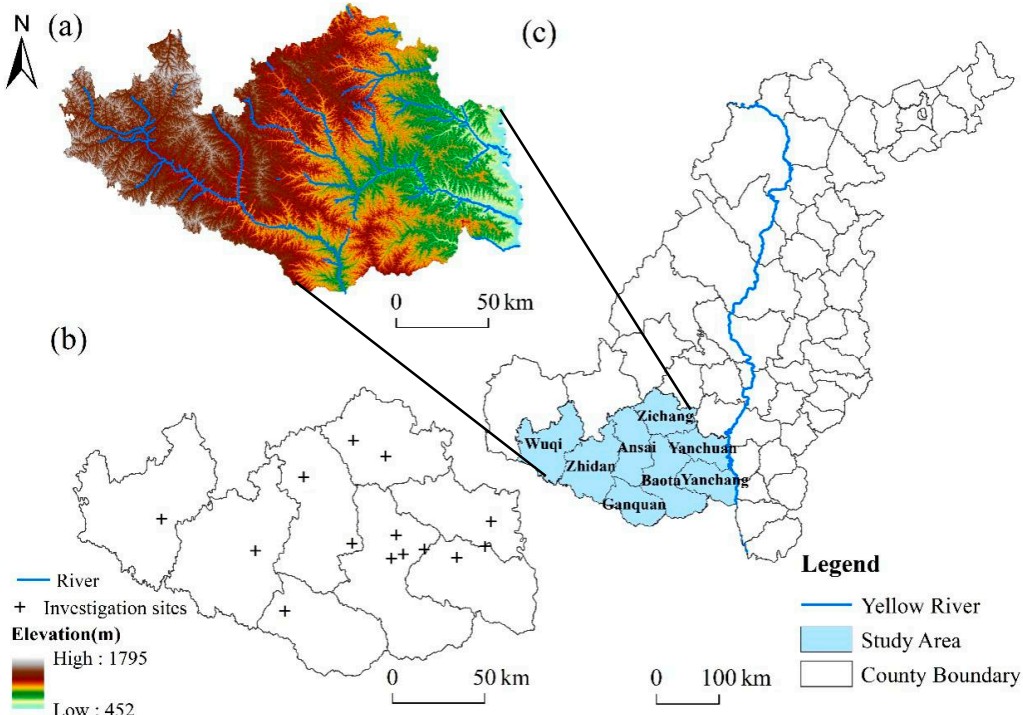

**Figure 2.** Location and investigation sites of the study area. (**a**) shows the digital elevation map of the study area, (**b**) shows the distribution of investigation sites in the study area, and (**c**) shows the loess hilly and gully region.

### 3.2. Data Sources and Processing

The data used in this study are mainly divided into two categories: remote sensing data and socio-economic data. The socio-economic data were mainly obtained through field surveys, field interviews, and statistical yearbooks. Field interviews mainly include four dimensions: natural ecological dimension, farmers' livelihood source and improvement dimension, agricultural planting and development dimension, ecological landscape, and environmental safety dimension. The remote sensing data originated from the United States Geological Survey (USGS) during 1995–2018 (http://glovis.usgs.gov/) and Landsat image data with a spatial resolution of 30 m from 1995 to 2018 were selected as the basic data for extracting gully farmland. The land use types were interpreted by referring to the classification method of CAS's resource and environment information database in 2018 (http://www.resdc.cn/). Besides, the crop vegetation grew luxuriantly from April to October, and the identification accuracy of the image was higher than in other months. After testing, the accuracy of the land-use types was over 85%.

The 30 m resolution digital elevation data in 2018 (http://www.gscloud.cn/), Google Earth images and high-resolution land classification data were used as training samples and verification basic data. Additionally, the selected vector road network data come from Open Street Map website in 2018 (http://www.openstreetmap.org), and the socio-economic data come from the Yan'an Statistical Yearbook and field investigation, and the socio-economic data was spatialized based on ArcGIS 10.4.

### 3.3. Research Methods

### 3.3.1. CART Decision Tree Algorithm

The classification methods such as artificial neural networks (ANN), decision trees (DT), and support vector machines (SVM) are widely used in the classification of remote sensing images [30]. Among them, the decision tree classification method can effectively excavate the spectral characteristics of the image and can solve the problem of the overlap of the remote sensing image spectrum to a greater extent. The commonly used algorithms in the DT decision tree classification method are: C4.5, CART, ID3, etc. [31], and the CART algorithm uses the Gini coefficient of the balanced distribution of income in economics as the criterion for determining the optimal test variable. The formula is as follows:

$$\text{Gini (D)} = 1 - \sum_{i=1}^{n} p_i^2 \tag{1}$$

In the formula, D is the data set, $n$ is the classification number, and $p_i$ is the distribution probability.

Compared with other decision trees, the model of the CART decision tree is simple. The classification threshold is determined by training samples, and the decision tree is automatically established (Figure 3). Therefore, it is less affected by other factors, and the recognition accuracy is higher.

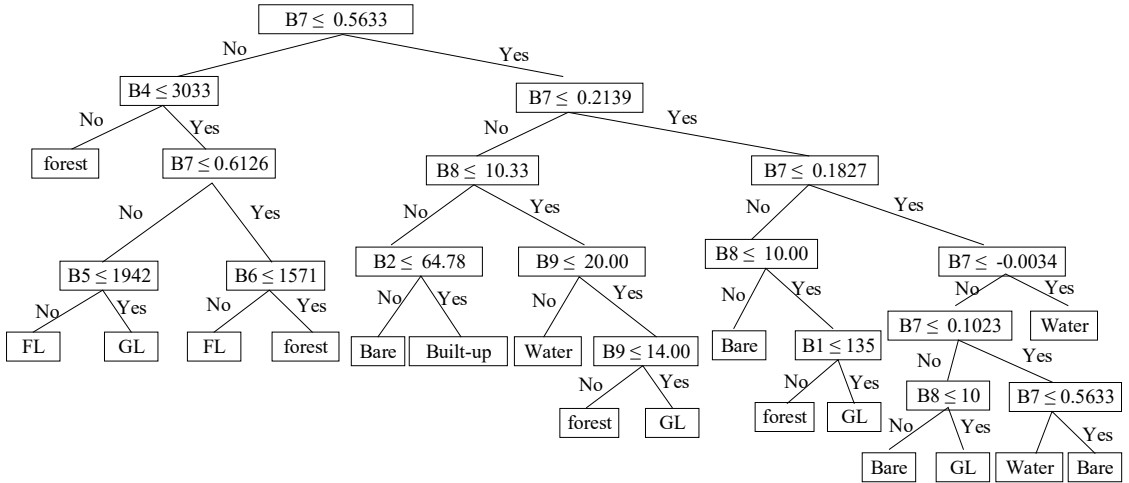

**Figure 3.** Decision tree classification process. Note: B1–B8 respectively correspond to the 8 bands of the composite image.

### 3.3.2. Gully Farmland Classification

Gully farmland can also be called "dam land" (Figure 4), which is a new form of agricultural production land of different types (Table 1) [32]. The identification rule of gully farmland is as follows: Step 1: The range of farmland in 2018 is extracted as the background data of gully farmland, and then the pixel (T) of other years is obtained based on the change of the range of the farmland. Besides, the suspected gully farmland range with the specific research period (1995–2018) is obtained based on the above judgment. Step 2: Based on the range of gully farmland in year (t-1), and the overlapping part of the suspected gully farmland in year (t-2) and the suspected gully farmland in year (t-1) is extracted again. This part is judged to belong to the range of gully farmland in year (t-2). Step 3: According to this method, the range of gully farmland in a continuous sequence of prescribed years is obtained.

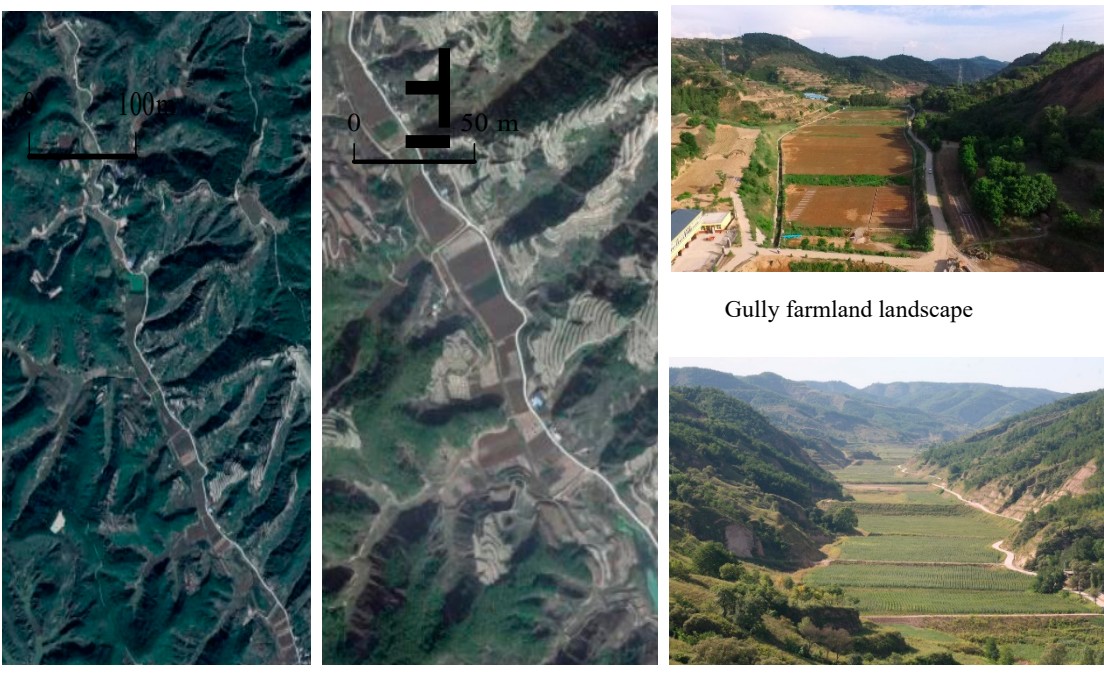

Typical gully remote sensing image      Local remote sensing image      The overall landscape

**Figure 4.** Gully farmland landscape in loess hilly and gully region.

**Table 1.** Reference standards of GUFL types.

| GUFL Types | Identification Standard | Sources | Interpretation Reference |
|---|---|---|---|
| Type I FFZ | The EFZs are neatly concentrated with darker colors and patterns, the rationale is dotted and the individual is clear. | QuickBrid (0.48 m) |  |
| Type II GV | There are certain roads and buildings around, the GVs regular rectangles with the same width, high reflectivity can be distinguished. | |  |
| Type III PF | The PF individuals are relatively regular polygons and the colors are mainly dark green. | |  |
| Type IV GP | The GPs are green in the growing season, and the rest are yellowish-brown in strips. | |  |
| Type V ML | The EFs are distributed in strips, the individuals show ladder-like shape, and the single row is a large width. | |  |
| Type VI T | There are signs of consolidation in cultivated land, T individuals are distributed in strips, with fine texture and narrow width. | |  |

Note: Fruit forest zones (FFZ); Greenhouse vegetables (GV); Pond farming (PF); Grain planting (GP); mulberry leaves (ML); tobacco (T).

### 3.4. Modified Binary Logistic Regression Model

Binary logic regression is a valuable tool to analyze the relationships between dependent variables and independent variables, which is widely used in the analysis of land use driving forces [33–35]. Furthermore, spatial autocorrelation between land use transfer probability and land use types with adjacent grids couples to modify the deficiency of simple logical regression analysis. Comprehensively considering the spatial representation and driving factors difference and combining with the actual situation in the LHGR, 21 indicators of four driving factors are selected. According to the relevant research experience and the actual situation of the development in the LHGR, this paper attempts to comprehensively select the factors that affect the development of gully agriculture from different dimensions aspects: nature, social economy, and humanities. Among them, including socio-economic (SE), hydrothermal condition (HC), natural background (NB), and location condition (LC) (Table 2). Overall, the analytical principle of the driving mechanism is to explore the main driving factors of gully farmland change under the existing social-ecological background, and the time limit of each index is 2018.

**Table 2.** The driving index of GAPT.

| Indicator Types | Indicator Names | Unit |
|---|---|---|
| Socio-economic (SE) | Population density (POP $T_1$) | People/km |
| | Gross domestic product (GDP $T_2$) | Yuan |
| | Main roads density (MRD $T_3$) | 1/km |
| | Primary industry employment rate (PIER $T_4$) | % |
| | Urbanization rate (UR $T_5$) | % |
| | Per capita fiscal revenue (PCFR $T_6$) | Yuan/People |
| | Primary production change rate (PPCR $T_7$) | % |
| Hydrothermal condition (HC) | Mean annual temperature (MAT $T_8$) | °C |
| | Average annual precipitation (AAP $T_9$) | mm |
| | Accumulated annual temperature (AAT $T_{10}$) | °C |
| Natural background (NB) | Elevation (ELE $T_{11}$) | m |
| | Slope (SLOP $T_{12}$) | ° |
| | Terrain relief (TR $T_{13}$) | 1 |
| Location condition (LC) | Distance to county cities (DTC $T_{14}$) | km |
| | Distance to township (DTT $T_{15}$) | |
| | Distance to national road (DTNR $T_{16}$) | km |
| | Distance to main highways (DTMH $T_{17}$) | km |
| | Distance to provincial road (DTPR $T_{18}$) | km |
| | Distance to county road (DTCR $T_{19}$) | km |
| | Distance to main railways (DTMR $T_{20}$) | km |
| | Distance to river (DTR $T_{21}$) | km |

*3.5. Geographically and Temporally Weighted Regression Model*

Unlike the widely used Geographically and temporally weighted regression (GWR) model, which only takes spatial variation into account when estimating an empirical relationship [36], GTWR captures spatio-temporal heterogeneity based on a weighting matrix referencing both spatial and temporal dimensions. In this study, a GTWR model was fitted using the following structure:

$$Y_i = \beta_0\left(X_i^t, Y_i^t, T_i\right) + \sum_m \beta_m\left(X_i^t, Y_i^t, T_i\right) X_{im} + \varepsilon_i \tag{2}$$

where: $Y_i$ is the value of monitoring site $i$; $(X_i, Y_i)$ represents the central coordinates of monitoring site $i$; $\beta_0$ denotes the intercepts at a specific location $(X_i, Y_i)$ and year $T$; and $\beta_1 - \beta_m$ are the location-time specific slopes for 1-m, respectively. $\varepsilon_i$ is the error term for sample $i$.

## 4. Results

*4.1. Spatial Distribution Characteristics of GAPT*

The center of gravity coordinate migration of gully farmland in 1995–2018 is shown in Figure 5. From 1995 to 2000, the migration direction of the gully farmland in the study area tended to the northwest of the region as a whole. After 2000, the center of gravity of the gully farmland showed a divergent distribution trend of migration to the surroundings. In the past 20 years, the gravity center of gully farmland moved northward, among which, Wuqi County, Baota County, and Ansai County have high consistency in migration direction. The migration direction of gully farmland in Baota County is always northward, while Wuqi County and Ansai County move southeast. The spatial conversion intensity of the gully farmland in Ganquan County and other counties is relatively intense, and the migration direction remains unchanged before 2005, and gradually expands to the opposite direction after 2005. In other counties, the direction of change in the stages of gully farmland is divergent, and the overall characteristics are relatively insignificant (Figure 5).

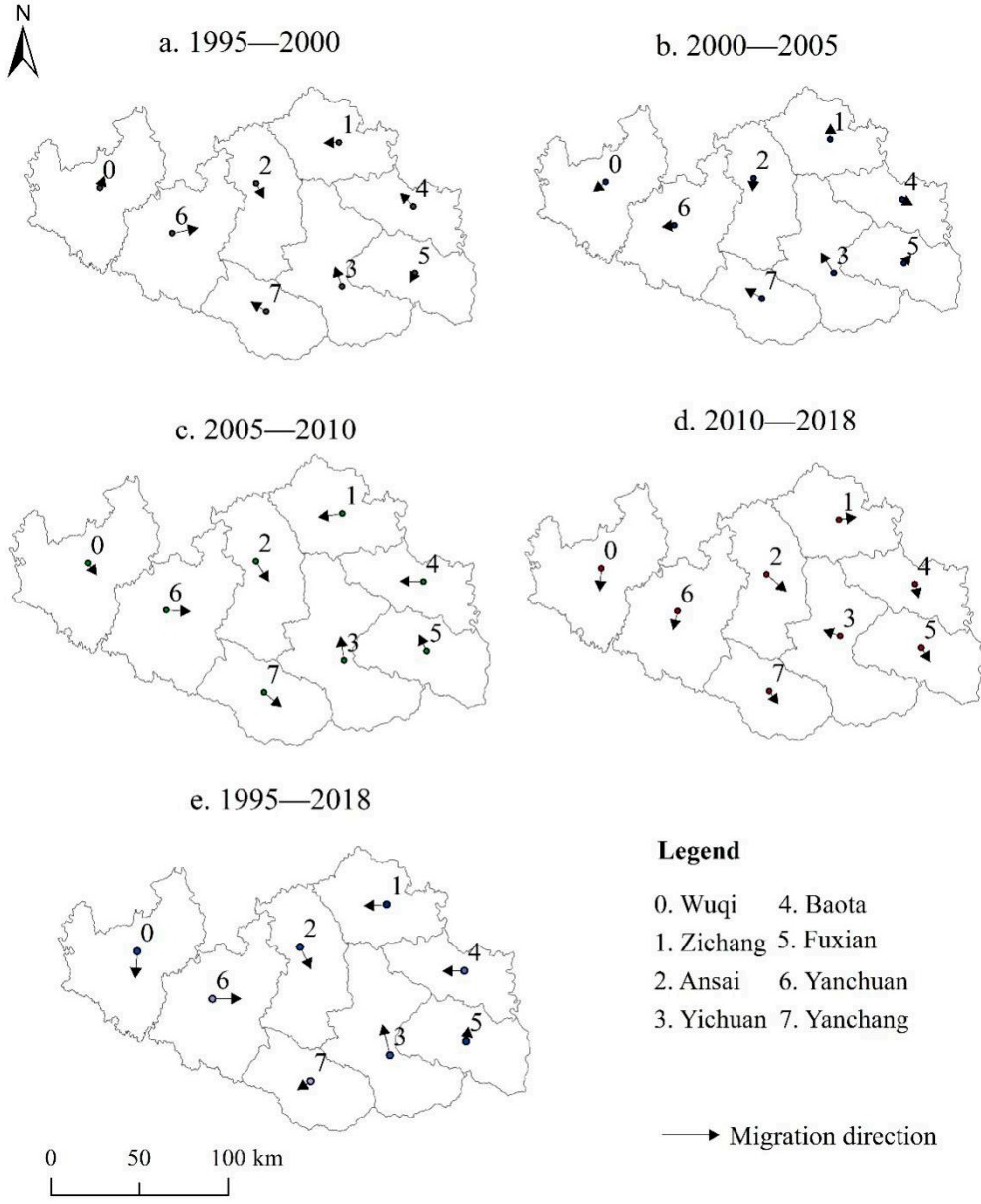

**Figure 5.** The center of gravity coordinate migration of gully farmland in districts of Yan'an City.

The spatial distribution characteristics of different types of gully farmland are shown in Figure 6. The growth of the gully farmland is mainly distributed in the relatively gentle low-altitude flat dam area, including elevation (700–1300 m), terrain relief (40–180 m), and slope (0–20°). The growth of oilseeds (IV), mulberry leaves (V), and tobacco leaves (VI) were negatively correlated with the distance between rivers, county roads, and provincial roads, and the increase was mainly within 5 km from rivers and 18 km from provincial roads. In general, the influence of the regional centers of townships on the cultivation of gully farmland was more significant than that of the regional centers of districts and counties. There was a strong market location orientation between economic fruit forest (I) and greenhouse vegetables (II), which was more manifested as agglomeration within the township, showing a "bottom-up" market location-oriented law.

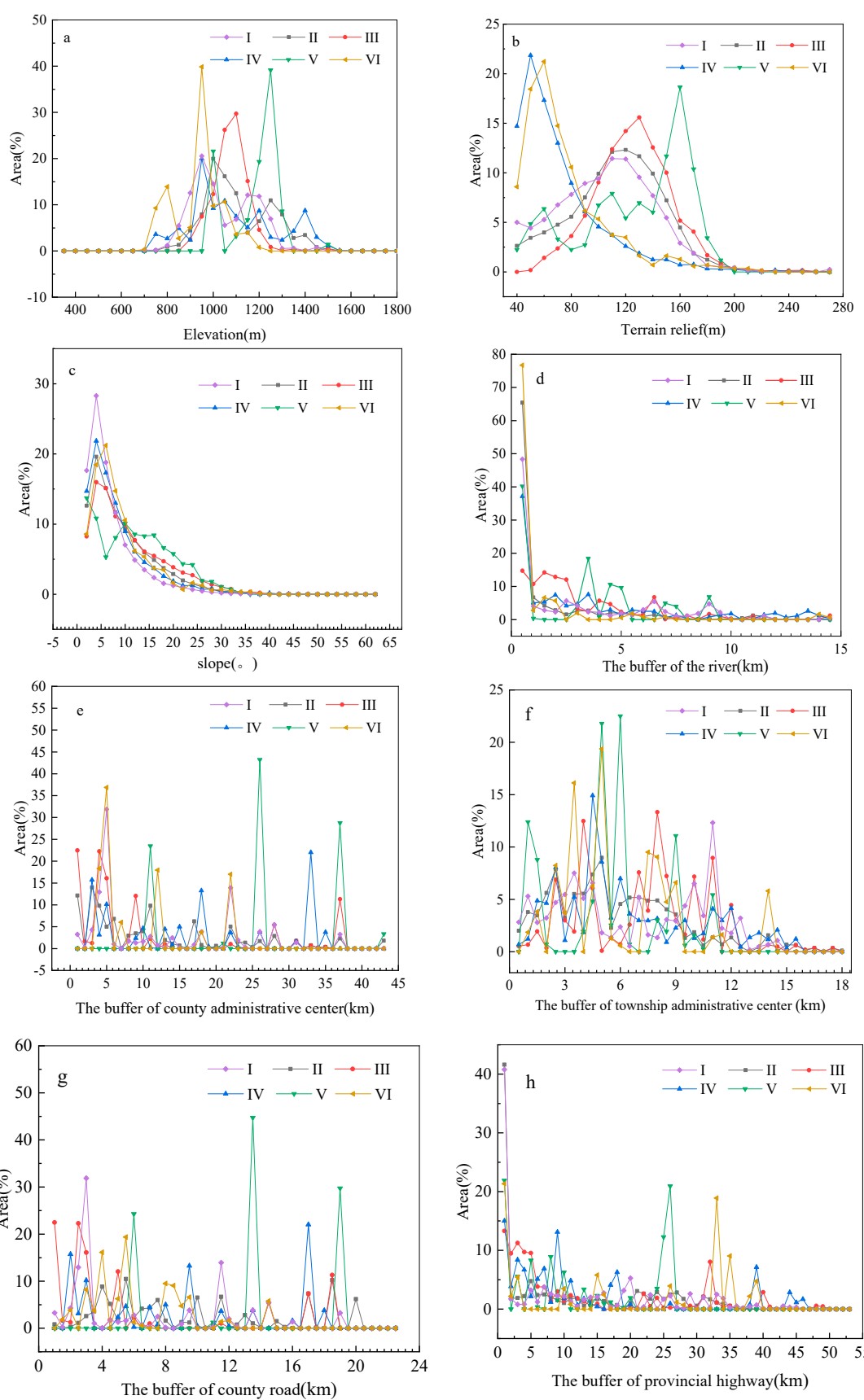

**Figure 6.** Spatial variation of gully farmland in Yan'an City. Note: Fruit forest zones (I); Greenhouse vegetables (II); Pond farming (III); Grain planting (IV); mulberry leaves (V); tobacco (VI).

### 4.2. Analysis of the Main Influencing Factors on GAPT

Drawing on the enlightenment of existing research and the relevant methods [37–39], the gully farmland expansion region (1), contraction region (−1), and no obvious change region with a total of 4861 in the four periods of the study area were analyzed in ArcGIS 10.4. The numbers of the three types of regions are roughly equal to ensure the stability of the coefficients of the explanatory variables in the model. During the study period, the sample data were processed by GTWR, and all passed the significance test ($p < 0.05$) (Table 3). Moreover, a linear regression OLS model was used to verify the applicability of GTWR. It could be seen that the $R^2$ value of the GTWR model was higher than that of the OLS model, and the Akachi information criterion (AIC) index was lower than that of the OLS model (Table 4). AIC was an important indicator of good model fitting, and the smaller the value, the higher the accuracy [35]. The AIC value (11.38/11.54 = 98.61%) and $R^2$ value (0.22/0.14 = 157.14%) of the GTWR model were significantly higher than OLS. Since the GTWR model added a time dimension, it was higher than that of the traditional model in dealing with spatio-temporal non-stationary data.

**Table 3.** GTWR parameter estimate summaries.

| Indicators | Minimum Value | 1/4 Quantile Value | Median Value | 3/4 Quantile Value | Maximum Value |
|---|---|---|---|---|---|
| Intercept C | −0.3919 | −0.0565 | 0.0257 | 0.1320 | 0.6239 |
| T6 | −1.3399 | 0.0229 | 0.1001 | 0.2295 | 0.5488 |
| T11 | −2.2942 | −0.4253 | −0.1709 | 0.0469 | 0.8249 |
| T15 | −3.8430 | −0.1501 | 0.1863 | 0.4926 | 1.7699 |
| T14 | −6.6169 | −0.6932 | −0.2322 | 0.4463 | 1.5808 |
| T15 | −5.1133 | −0.4521 | −0.0995 | 0.2985 | 1.5287 |
| T19 | −8.7846 | −0.3760 | −0.1529 | 0.0356 | 0.8274 |
| T20 | −1.0248 | −0.2259 | 0.0695 | 0.3779 | 3.8356 |
| T18 | −1.7000 | −0.4558 | 0.3068 | 0.9633 | 1.0030 |
| T9 | −2.6693 | −2.3905 | −1.0851 | −0.2395 | 1.2369 |
| T16 | −6.1140 | −2.3399 | −0.3992 | 0.5243 | 0.7178 |
| T17 | −4.1616 | −0.3849 | 0.2826 | 0.9404 | 2.8666 |
| T10 | −1.4554 | 2.7472 | 4.6608 | 8.6157 | 5.8123 |

**Table 4.** Comparison of model diagnostic results.

| Models | Correlation | AIC | $R^2$ | $F (r^2)$ | $p (r^2)$ |
|---|---|---|---|---|---|
| GTWR | 0.525 | 11.3803 | 0.226 | 5.546 | <0.001 |
| OLS | 0.377 | 11.5464 | 0.140 | 56.311 | <0.001 |

Comprehensively considering the spatial representation and driving factors difference and combining with the actual situation in gully areas [40,41], 21 indicators of four driving factors were selected, including nature, economy, population, transportation, location, and policy. To eliminate the influence of multi-factor collinearity, this paper firstly analyzed the correlation of the driving factor set, and the result showed that the correlation of selected factors was significant. Then, principal component analysis was performed on the driving factor set, and the load matrix and coefficient matrix of each component were obtained. Moreover, regression analysis was used to obtain the comprehensive score value of each sample. The dominant factors of driving variables are shown in Table 5.

**Table 5.** Dominant factors of principal component driving variables.

| Classification | Principal Component | Composition | Dominant Direction | Driving Type |
|---|---|---|---|---|
| Continuity | $F_1$ | $T_2$, $T_3$, $T_4$ | Social and economic development | Multivariate |
| | $F_2$ | $T_6$, $T_2$ | Investment and development dominance | Dual factor |
| Periodicity | $F_3$ | $T_{20}$, $T_{17}$, $T_1$ | Location dominance | Multivariate |
| | $F_4$ | $T_{14}$, $T_{16}$ | Traffic dominance | Dual factor |
| | $F_5$ | $T_2$, $T_4$, $T_{19}$ | Economic dominance | Multivariate |
| | $F_6$ | $T_{11}$, $T_{13}$ | Terrain slope dominance | Dual factor |
| | $F_7$ | $T_{14}$, $T_{17}$ | Traffic dominance | Dual factor |
| | $F_8$ | $T_{13}$ | Topographic relief dominance | Single factor |
| | $F_9$ | $T_1$ | Population density dominance | Single factor |
| | $F_{10}$ | $T_{21}$ | Water factor dominance | Single factor |

*4.3. Spatio-Temporal Differentiation of Influencing Factors of GAPT*

According to Figure 7, it can be seen that the high value of $F_1$ in the study area during the 20 years has shifted from west to east first, and then expanded from south to north. The driving difference is reduced and gradually tends to be balanced, eventually forming a clustering and distribution situation around the Baota district. The positive high value from 1995 to 2000 was concentrated in Yanchang-Ansai-Wuqi, while Zichang, Yanchuan, and the central area of Baota were relatively low value. From 2000 to 2005, the negative high value shifted to Zhidan, and the positive and negative effect intensity was significantly different between northwest and southeast. The positive high values from 2005 to 2010 were mainly distributed in Ganquan and Ansai, and the difference in the driving factor value range narrowed and tended to be balanced. The pattern evolution from 2010 to 2018 gradually formed a circle with the Baota as the core, the positive high value of Ganquan, and the negative low value of Yichuan. From the perspective of the evolution of economic factors, the economic development of the Baota has a higher spillover effect on the expansion of gully farmland. The location dominant factors ($F_3$) were quite different, and the positive effect of the township was stronger than that of county administrative centers in which population and economy were the two most significant factors. The population was also the direct driving force for the conversion of gully farmland, and economic development was the core driving force. The location characteristics directly affected the population flow and capital flow and promoted the transformation of gully farmland. In addition, some related policies (Grain for Green project) and engineering measures (Gully Land Consolidation) had accelerated the conversion of gully farmland, thereby promoting the spatial transformation of gully farmland.

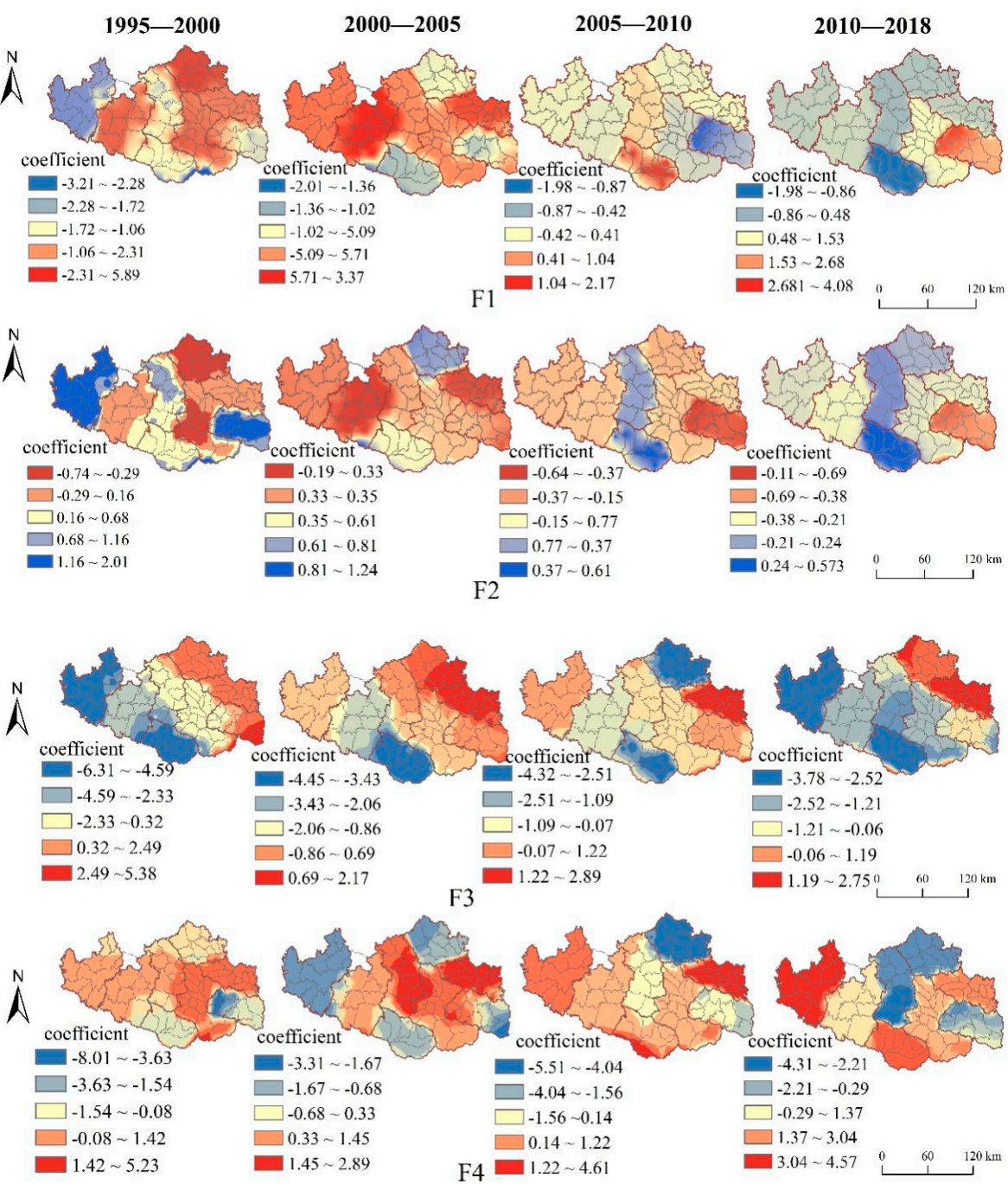

**Figure 7.** *Cont.*

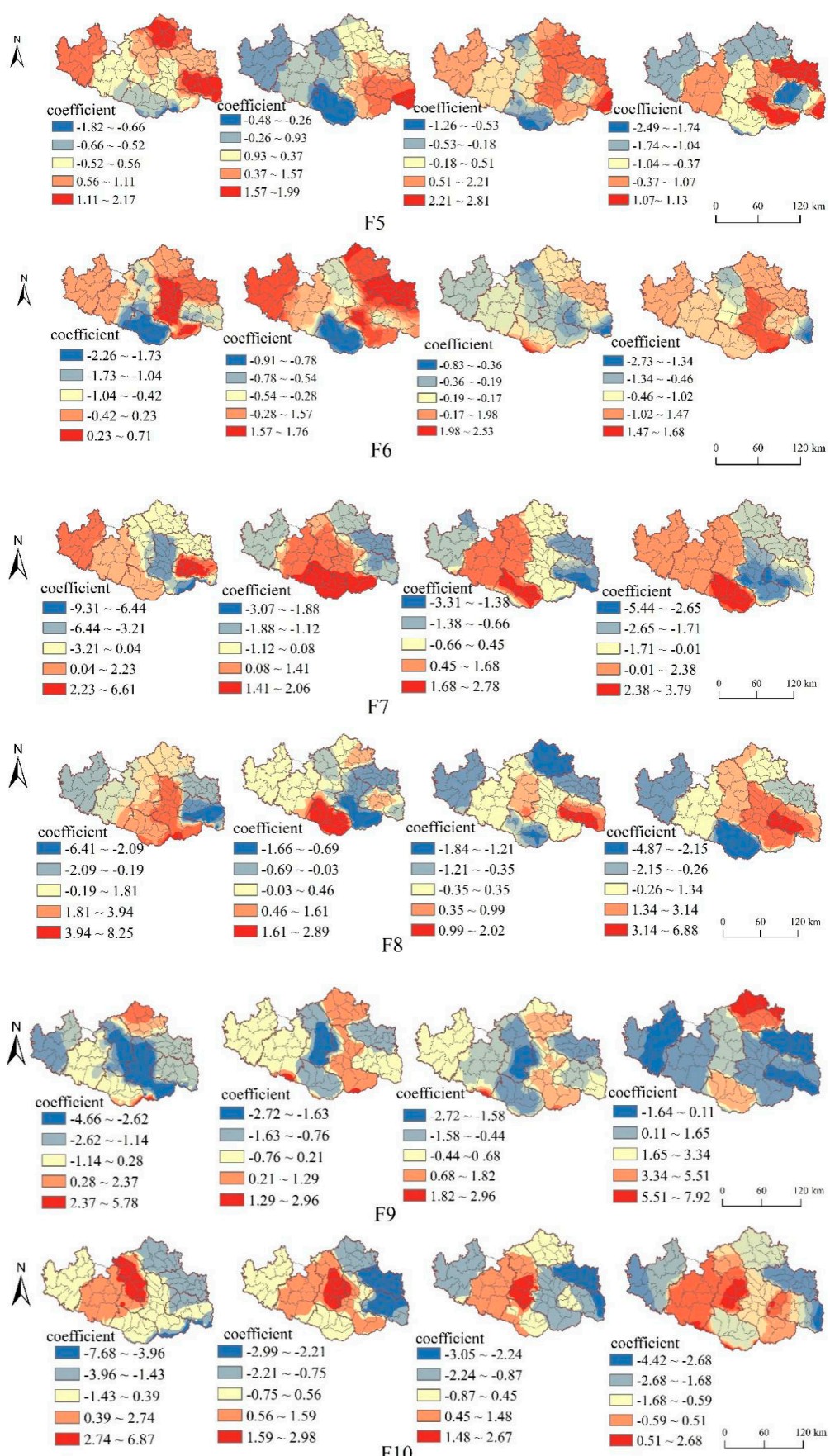

**Figure 7.** Spatio-temporal variation pattern of dominant factor coefficient of gully farmland expansion.

According to Figure 7, the changes of $F_3$ and $F_5$ were relatively similar. Normally, natural factors had no significant effect on gully farmland conversion from 1995 to 2018, and the overall north-south changes in space were obvious, mainly due to the difference in topography and geomorphology in the north and south of the study area. According to $F_4$ (traffic location dominant), the influence intensity distribution of distance to national road and distance to main highways presents two significant characteristics: First, it formed a distribution pattern of high in the south and low in the north at the regional level, and the high-value region gradually shifted from the southwest to the east of Yan'an City over time. Second, some high-value regions were scattered in some county boundaries, such as the southwest of Yanchang near the border of Baota District, the northwest border of Yanchuan, the southwest part of Baota, and the border area of Ansai and Zhidan.

The distance to the provincial road ($F_4$) had an overall evolutionary trend toward being high in the central area and low in the surrounding area. From 2000 to 2005, the vertical expansion to Baota district was a "W"-shaped high-value area layout. From 2005 to 2018, it further expanded to the north to Ansai County, and the positive effect on the expansion of gully farmland in the Baota area was further enhanced. Water dominated factor ($F_{10}$) was mainly distributed in Wuqi County and Yanchang County along Luo River and Yanhe River from 1995 to 2000. The high-value area gradually moved down from 2000 to 2005. Since 2005, the scope of high-value areas was further expanded, showing a distributed pattern of dots in the area.

From the perspective of the driving force change range of gully farmland conversion, the largest growth of driving factor was the economic factor of per land GDP. From the perspective of the regional change pattern of the driving force, various geomorphic factors have different effects on the expansion of gully farmland, which was closely related to the geomorphological differentiation characteristics of the gully area. The impact of economic development on gully farmland presents a "core edge" funnel-shaped distribution, and its changes tended to "flow" to the core. The influence of the neighboring factors on the roads in Yan'an had its own characteristics, which was closely related to the spatial distribution of the road network. Areas with significant changes were concentrated in the central area of Yan'an City and focused on the Baota area, roughly forming a "T"-shaped distribution pattern.

### 4.4. Analysis of GAPT Response Mechanism

The factor coefficients of the four stages in the study area from 1995–2000, 2000–2005, 2005–2010, and 2010–2018 were statistically separated, and the average factor coefficients of each stage were obtained as shown in Figure 8.

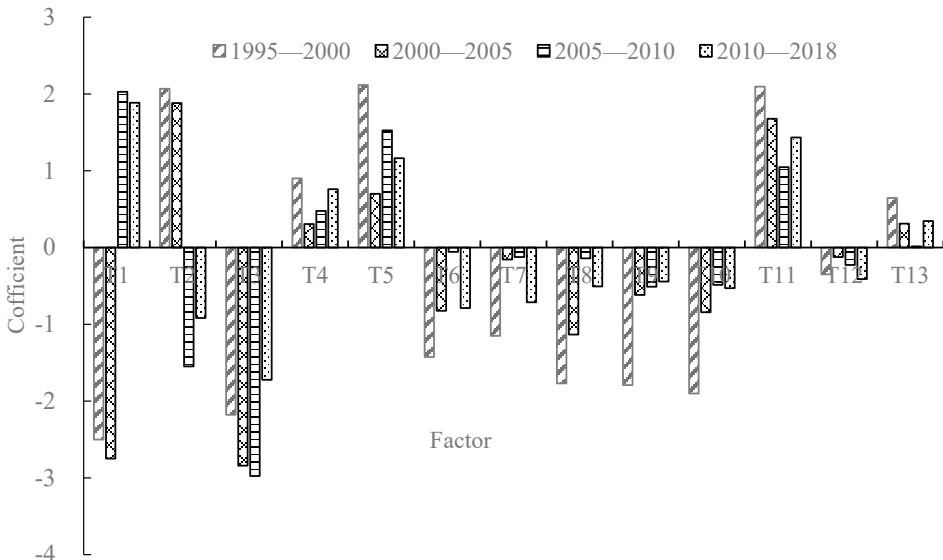

**Figure 8.** Time series distribution of factor average coefficient change in each period.

(1) Judgment of the whole area core factors: From 1995 to 2018, the conversion of gully farmland of the study area was mainly based on the population and GDP per land area of socio-economic factors. The slope of natural factors and the distance from the water area of neighboring factors were the core influencing factors. Among them, the population factor and slope factor had always been in a dominant position, and they were the main driving factors for the conversion of gully farmland. The intensity of population was higher than that of the slope during 1995–2000 and 2010–2018. The driving effect of the slope on gully farmland conversion during 2000–2005 and 2005–2010 was obvious.

(2) Temporal evolution of the whole area core factors: Slope was the most influential factor in the conversion of gully farmland. The intensity of its effect was a fluctuating and rising trend from the beginning of 1995 to 2010, and the intensity was the greatest in 2010. The relationship between population and the transformation of gully farmland was negatively correlated in the first 10 years. The overall effect of the population on the expansion and transformation of gully farmland showed a downward trend. Slope aspect and topographic relief factors were positively correlated with gully farmland shrinkage. From the whole study period, the intensity of aspect effect intensity first decreased and then increased, and remained flat. The influence of traffic neighborhood factors generally showed a slight decrease. The gully was negatively correlated with the transformation of farmland shrinkage. The distance from the center of the county was negatively correlated with the shrinkage of gully farmland. The intensity of the impact first decreased and then increased, showing an overall increasing trend. There was a positive correlation between the distance from the water area and the center of town and the change of gully farmland shrinkage in Yan'an City, and the intensity of these factors was gradually declining.

At the county level, it was the basis of regional differentiation governance to identify the driving force of gully farmland contraction. In order to identify the difference in the conversion drive of the gully farmland between the counties and determine the time series change of the drive strength, the incidence of factors driving the transformation of gully farmland contraction was compared among the counties as follows (Figure 9). Among these, the positively related variables were directly taken as logarithm exp (g), and negatively related variables were converted into positive cumulative values through the factor of 1/exp (g).

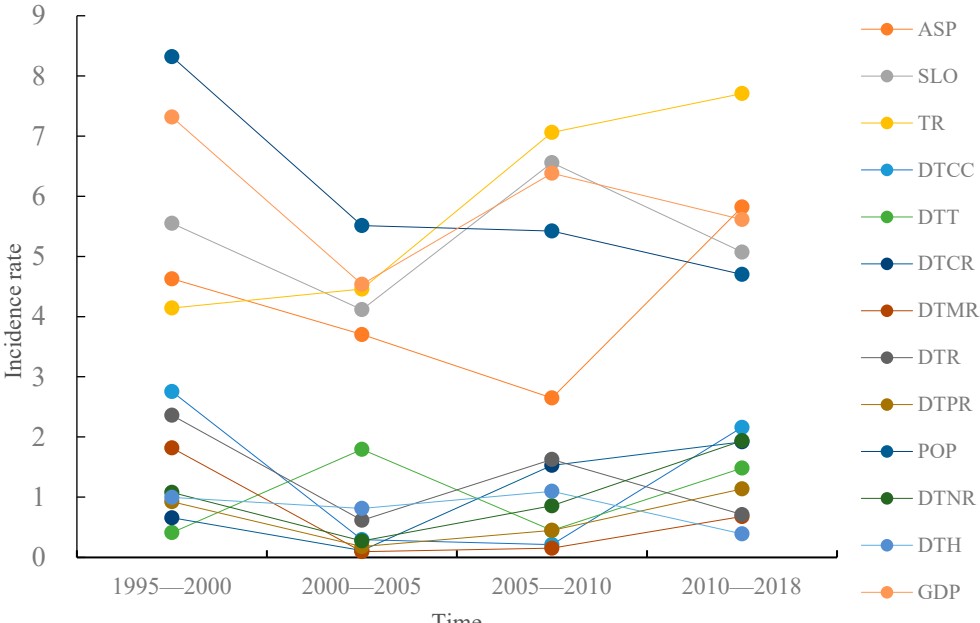

**Figure 9.** The positive incidence rate of GAPT.

The core driving factors for the conversion of gully agricultural land in the past 20 years were population, topographic relief, slope, aspect, and GDP per land. The first driving factor of Yanchang and Zhidan was population. The per land GDP in Baota was the primary driving factor before 2000, and then population became the primary driving factor after 2000. Moreover, Ganquan, Ansai, and other counties showed obvious phased characteristics. In regional spatial distribution, the counties with strong population and economic driving force were concentrated in the east and middle of Yan'an. Most of the counties with strong driving forces such as slope, aspect, and topographic relief were in the west. There were significant differences between the eastern and western regions at the county level. From the perspective of the time sequence of the slope of the dominant factor affecting the conversion of gully farmland. Except for Ganquan, the driving force for the conversion of gully farmland by the slope of the counties had not declined. It showed that the restriction of natural background conditions was still strong.

## 5. Summary and Implications

### 5.1. General Law of Rural Land Use Evolution in Gully Areas

Rural development gets a lot of attention around the world. However, compared with cities, rural areas are backward for reasons that include migrant mobility, poverty, labor quality, biased policy, and week rural land management [42]. Generally, the cultivated farmland within the rural man-land system in plain areas is evenly distributed [43], and the effect of intensive cultivation and utilization is significant, which is easier to integrate the land resources than that in hilly areas.

Hilly rural settlements and agricultural land distribute scattered, and the tillage radius is wider than that in the plain. The unique geographical characteristics shape the unique rural man land system to a certain extent. In addition, environmental elements are the essential factors influencing spatial distribution [44]. The natural environment, including temperature, precipitation, terrain, and rivers, is the basis for the formation and development of agriculture and the countryside [45]. The farming environment created by conscious social labor in the process of agricultural production has a significant impact on its development and distribution [46]. In the socioeconomic environment, dynamic changes in the spatial pattern of GAPT are determined by factors of urbanization rate, main highway density, per land GDP, the proportion of primary industry employment and land consolidation planning, etc.

Moreover, the influence of the commodity economic location environment on rural land use and development continues to increasing [47]. Gully agriculture tends to develop in the win-win situation of ecology and economy. GAPT is comprehensively affected by a variety of internal and external factors, which not only depend on the external resource and environmental status of the gully, but also have a comprehensive effect on the gully's rural internal economy, society, ecology, and culture. On the micro-valley level, GAPT is related to the benefits and efficiency of exchanging energy. On the macro-regional level, GAPT is a summary and generalization of the spatial differentiation of gully regional production-living-ecology.

This paper represents the general law of gully rural land use in LHGR, and the process of GAPT can reveal the LUT mode in China's loess plateau and modern gully man-land system development in LHGR (Figure 10). With rural population outflow, GAPT will promote the gully living space reconstruction and intensive use of production, and the man-land system development tends to be coordinated and efficient. Specifically, GAPT reflects the characteristic of gully agriculture crops, ecological maintenance, and rural living environment construction in LHGR. The combination accelerates rural transformation in LHGR, which has an indicative implication for the high-quality development of agriculture and rural in China's Yellow River Basin.

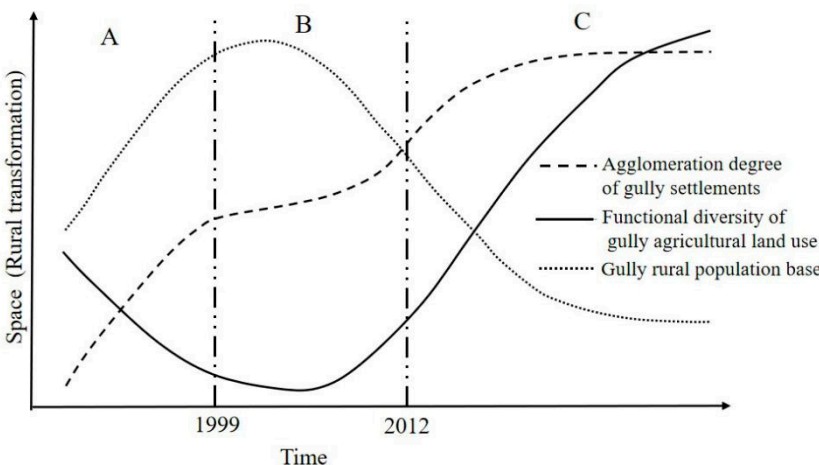

**Figure 10.** The transformation model of gully rural.

*5.2. Policy and Practical Implications of Gully Agriculture Development*

According to the characteristics of GAPT types, rational rural land use and agricultural planning can improve the gully agricultural production environment by enhancing the conversion rate of GAPT and improving regional infrastructure supporting engineering. Policymakers should establish adaptive measures for gully land consolidation and formulate structural readjustment measures for different gully agricultural planting types to promote intensive land use.

The phenomenon of GAPT in the LHGR reveals the general law of the evolution of the river basin and rural man-land system driven by ecological construction and gully land consolidation engineering, that is, ecological security and food security. In the new era, the gully agriculture of the Loess Plateau should be based on the economic-ecological "win-win" benefits, focusing on ecological civilization and green sustainable development, and using the two major projects—the Grain for Green project and Gully Land Consolidation—as important platforms for transformation and development, and coordinating watersheds up-and-down collaboration, gully and slope collaboration, multi-scale classification and coordination, giving full play to the technical support role of geographic engineering, and in-depth exploration of new ways to optimize gully agricultural production methods and innovate business management models. As the goal, the ecological economy is "win-win" as the direction, and modern geographic engineering is the technical means to finally achieve high-quality development of quality benefits and development efficiency [48]. GAPT in the LHGR has promoted the integration of policies such as the Grain for Green project and Gully Land Consolidation. On the premise of protecting and improving farmers' livelihoods, it is important to promote the integration of the three beings (production, living, ecological) in the gully region and bring to light the significance of promoting the integration of the three industries and high-quality development of the Loess Plateau.

*5.3. Limitations and Prospects*

There are still some shortcomings. For data and technical methods, this paper uses remote sensing image data from April to October to interpret the types of gully agriculture, and the crop and grassland are easy to distinguish. There may be some misclassification conditions. In addition, the spatial-temporal geographic weighted regression model realizes a comprehensive measurement in the spatial-temporal perspective and more precisely describes the temporal and spatial evolution characteristics of the driving factors and the dominant driving mechanism. However, how to improve the accuracy of image classification based on the spatial location information of ground objects and further optimize the "circle-belt-region" multi-level spatial structure system of human land system in rural areas of loess hilly and gully region [5,49], still need to be further explored.

It is undeniable that GAPT has significant positive economic benefits to gully rural areas, but whether it can bring ecological benefits is still unknown. In addition, policy as a non-quantitative factor is essential and crucial in GAPT. With the development of gully land consolidation engineering, the amount of gully farmland is also changing. At the same time, the willingness of farmers and the reform of property rights are also factors to be considered. Furthermore, by selecting typical types of areas, the process and mechanism of the changes of different types of regions are explored from the micro-scale, and then the scientific principles of the rural man land system based on gully farmland and the mechanism of rural "human-geosphere" are revealed should also be the focus of future work.

## 6. Conclusions

Under the background of "Grain for Green" (GFG) land management policies and the rural population transfer, the focus of gully farmland distribution gradually migrates down to low-altitude flat dam areas. Spatio-temporal distribution of different driving factor coefficients of GAPT has different effects. The intensity of the population and slope is always in the dominant position. The high-value area of GDP per land forms a funnel-shaped pattern of "core edge" in the northern and central-western regions, and its changes tend to be the core "flow". In the Loess Plateau, GAPT is driven by many factors, and the regional background differences and different actors will promote the development and contraction of gully farmland in different directions, but in principle, they are driven by the major national and regional development policies. In the development process, we should pay attention to the dual orientation of ecology and economy, maximize the sustainable use of gully farmland, explore the coordination and optimization of gully farmland and human settlements, and realize housing and industry synergy.

Current studies on gully agricultural production on the Loess Plateau based on gully land consolidation have good theoretical and practical significance. Taking a typical loess hilly and gully region of Yan'an City located in the center of the Loess Plateau as the case study area, this paper revealed the intensive land use under the background of population contraction in the Chinese Loess Plateau and its transformation trend by defining the gully agricultural production transformation (GAPT). To some extent, the multiple elements of the agricultural production system in gully areas from the perspective of farmer subject and regional production space and type change has been deconstructed towards the gully agricultural production development. However, more empirical studies in the gully region are still urgently needed to verify this key topic. In addition, we proposed a theoretical model for gully agricultural evolution in gully areas, this theoretical model needs further verification. It is worth mentioning that, we have evaluated the interaction mechanism that drives changes in the human–land relationship in the LHGR. We will further observe, monitor, and reevaluate the GAPT and report new progress in the near future.

Overall, the study area's typicality reflects the general law of rural land use evolution in gully areas. Policymakers can implement the adaptive measures for modern gully land consolidation measures and formulate structural readjustment measures for different gully agricultural planting types to promote intensive land use, ultimately achieving a balance of the regional modern rural man-land relationship.

**Author Contributions:** Conceptualization, Y.L.; Formal analysis, J.L. and L.Q.; Investigation, X.Z.; Y.L. and L.Q.; Software, Y.H.; Writing—original draft, L.Q.; Writing—review and editing, Y.L. and L.Q. All authors have read and agreed to the published version of the manuscript.

**Funding:** This research was funded by the Key Program of the National Key Research and Development Program of China (Grant 2017YFC0504701) and the National Natural Science Foundation of China (Grant No. 42101202, 41801172 and 42061037) and Fundamental Research Funds for the Central Universities (2021CDJSKJC02).

**Institutional Review Board Statement:** Not applicable.

**Informed Consent Statement:** Not applicable.

**Data Availability Statement:** Not applicable.

**Acknowledgments:** The insightful and constructive comments and suggestions from the anonymous reviewers are greatly appreciated.

**Conflicts of Interest:** The authors declare no conflict of interest.

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
