# Peer review of "Analysis of the Spatial Variations of Determinants of Gully Agricultural Production Transformation in the Chinese Loess Plateau and Its Policy Implications"

_land, doi:10.3390/land10090901_

Round 1
Reviewer 1 Report
Dear Authors,
Please check the spelling of Loess Hilly and Gully Region (LHGR) – whether it is capitals or not (e.g. see in abstract line 13), needs to be consistent, please check in the manuscrpit.
Figure 1. definitely requires further development – the source is not clear, time scale is unknown, quantity is also unknown.
Line 126 „Since returning farmland to forest in the LHGR...” please specify the date or at least mention the decade
Table 1. The time scale of data stops at 2018. In case it is possible, it would be necessary to further develop with newer data according to the evaluation and introduction of the recent state of the LHGR, as well.
Table 3. Considering the Social economy group of indicators, preferably use socio-economic category name instead. The NE category name needs to be rethink either. Please read through, check the indicator names (e.g. see urbanization „rat”) and correct them. According to the 5 indicator groups as drivers the explanation part is missing from the text. It seems to be quite subjective both in case the categories and also the indicators. It would be also important to describe the 5 groups and focus on correlation as well. Requires further explanation according to “Distance to…” or neighborhood type indicators whether the two selected indicators related to LC group are relevant in case of local conditions. Please explain why these indicators can not belong to the NE or even to category SE. Considering your assessment it would be interesting and has further added value to apply e.g. LISA evaluation.
Figure 12. see my comment as in case of Figure 1.
Conclusion part requires further development especially according to policy recommendations.
Author Response
Thanks a lot for your comments on our paper. We have revised our paper according to your comments:
1、Please check the spelling of Loess Hilly and Gully Region (LHGR) – whether it is capitals or not (e.g. see in abstract line 13), needs to be consistent, please check in the manuscrpit.
Answer: Thank you very much for the careful suggestion. We uniformly revise the spelling of loess hilly and gully region (LHGR) to ensure consistency in the manuscrpit.
2、Figure 1. definitely requires further development – the source is not clear, time scale is unknown, quantity is also unknown.
Answer: We quite agree with you, and revise all the manuscript including time scale 1999 and 2007. The revisions are in the manuscript.
3、Line 126, Since returning farmland to forest in the LHGR...” please specify the date or at least mention the decade.
Answer: Thank you very much for the question. We have added the time for project implementation to make the logical expression clearer. The revisions are in the manuscript “Since 1999 the Grain for Green Project (GGP) in the LHGR, especially the Gully Land Consolidation (GLC) project in 2012, the gully rural human–land relationship has changed subtly”.
4、Table 1. The time scale of data stops at 2018. In case it is possible, it would be necessary to further develop with newer data according to the evaluation and introduction of the recent state of the LHGR, as well.
Answer: We quite agree with you. The overall transformation of farmland on the Loess Plateau began in 2000. Since 1999, as the first pilot areas for the conversion of farmland to forest and gully control projects, the hilly and gully regions of the Loess Plateau took the lead in realizing the change in surface color from "yellow" to "green". The transformation of production space from "slope land" to "gully", so the starting point for this study was selected since 1995, and the social and economic kilometer grid data such as population and land per capita GDP from the Resource and Environmental Science Data Center of the Chinese Academy of Sciences were used. The study period covers the comparative analysis before and after the implementation of major policies on the Loess Plateau (Since 1999 the Grain for Green Project (GGP) in the LHGR, and the Gully Land Consolidation (GLC) project in 2012). We will further observe, monitor and reevaluate the GAPT and report new progress in the near future.
5、Table 3. Considering the Social economy group of indicators, preferably use socio-economic category name instead. The NE category name needs to be rethink either. Please read through, check the indicator names (e.g. see urbanization „rat”) and correct them. According to the 5 indicator groups as drivers the explanation part is missing from the text. It seems to be quite subjective both in case the categories and also the indicators. It would be also important to describe the 5 groups and focus on correlation as well. Requires further explanation according to “Distance to…” or neighborhood type indicators whether the two selected indicators related to LC group are relevant in case of local conditions. Please explain why these indicators can not belong to the NE or even to category SE. Considering your assessment it would be interesting and has further added value to apply e.g. LISA evaluation.
Answer: Thank you very much for the careful suggestion. According to the relevant research experience and the actual situation of the development in the loess hilly and gully region, we have read and checked the indicator names, and the indicators are re combed and systematically explained. This paper comprehensively selects the factors that affect the development of gully agriculture from different dimensions aspects: nature, social economy, and humanities. Among them, including social economy (SE), hydrothermal condition (HC), location condition (LC), natural background (NB) and neighborhood traffic location (NT).
Social economy (SE) indicators refer to the driving factors for the development of gully agriculture over time in the process of urbanization. Among them, economic income indicators such as fiscal revenue and GDP can reflect regional economic strength, and infrastructure such as roads can reflect economic investment and social development. In previous relevant studies, the population change are usually classified into socio-economic factors. Considering the particularity of the location conditions of gully hilly and mountainous areas, as well as the particularity of the Grain for Green Project and the Gully Land Consolidation project on the Loess Plateau, the "flow" relationship between the population and economic distribution in the study area is constantly changing, Therefore, it is summarized into socio-economic factors, and finally seven indicators are determined, such as urbanization rate, population density, proportion of primary industry employees, per capita financial income and primary industry change rate.
The economic location condition (LC) is an important engine of agricultural and rural development, which is characterized by the distance from the county administrative center and the township administrative center.
Table The driving index of GAPT.
|
Indicator types |
Indicator names |
Unit |
|
Social economy (SE) |
Population density (POP T1) |
People/km |
|
Gross domestic product(GDP T2) |
Yuan |
|
|
Main roads density(MRD T3) |
1/km |
|
|
Primary industry employment rate(PIER T4) |
% |
|
|
Urbanization rate(UR T5) |
% |
|
|
Per capita fiscal revenue(PCFR T6) |
Yuan/People |
|
|
Primary production change rate(PPCR T7) |
% |
|
|
Hydrothermal condition(HC) |
Mean annual temperature(MAT T8) |
℃ |
|
Average annual precipitation(AAP T9) |
mm |
|
|
Accumulated annual temperature(AAT T10) |
℃ |
|
|
Location condition (LC) |
Distance to county cities(DTC T11) |
km |
|
Distance to township (DTT T12) |
km |
|
|
Natural background (NB) |
Elevation(ELE T13) |
m |
|
Slope(SLO T14) |
。 |
|
|
Terrain relief(TR T15) |
1 |
|
|
Neighborhood traffic location(NT) |
Distance to national road(DTNR T16) |
km |
|
Distance to main highways(DTMH T17) |
km |
|
|
Distance to provincial road(DTPR T18) |
km |
|
|
Distance to county road(DTCR T19) |
km |
|
|
Distance to main railways(DTMR T20) |
km |
|
|
|
Distance to river(DTR T21) |
km |
Natural environment indicators refer to natural factors such as water, soil, light and environmental conditions such as slope and elevation that affect the growth of crops. Water and soil resources are the basis of agricultural production, temperature and light are the guarantee of agricultural production, and natural environments such as slope and elevation are also the conditions that restrict the development of agricultural production. Finally, six indicators such as regional water resources abundance, annual average temperature, slope and elevation are selected. Location conditions include not only the location in the sense of geographical space, that is, the traffic location in the general sense, but also the economic location that promotes economic development. Due to the limitations of their own site conditions and transportation, the location of hilly and mountainous areas often makes their population and economy constantly changing, which determines the spatial direction of "population flow, economic flow and information flow".
Neighborhood traffic location (NT) refers to the convenience of traffic and transportation. It is also an important bridge connecting agricultural villages and towns in the gully basin. The geographical location is characterized by five indicators, such as the distance to various types of highways and the distance to major rivers.
6、Figure 12. see my comment as in case of Figure 1.
Answer: We quite agree with you, and revise the manuscript. It is designed to display in time scale and quantity.
7、Conclusion part requires further development especially according to policy recommendations.
Answer: Thank you very much for the careful suggestion. According to policy recommendations, the conclusion part is further expanded. The revisions are in the manuscript.
Current studies on gully agricultural production on the Loess Plateau based on gully land consolidation has good theoretical and practical significance. Taken a typical loess hilly and gully region of Yan’an City located in the center of the Loess Plateau as case study area, this paper revealed the intensive land use under the background of population contraction in the Chinese Loess Plateau and its transformation trend by defining the gully agricultural production transformation (GAPT). To some extent, the multiple elements of the agricultural production system in gully areas from the perspective of farmer subject and regional production space and type change has been deconstructed towards the gully agricultural production development. However, more empirical studies in gully region are still urgently needed to verify this key topic. In addition, we proposed a theoretical model for gully agricultural evolution in gully areas, this theoretical model needs further verification. It is worth mentioning that, we have evaluated the interaction mechanism that drives changes in the human–land relationship in the LHGR. We will further observe, monitor and reevaluate the GAPT and report new progress in the near future.
We will try our best to revise the paper!
Thanks a lot again for having reviewed our manuscript.

Reviewer 2 Report
The authors of the study raised an important problem of the development of loess ravines. The work is written correctly. Sufficient methods have been applied. I think that the article brings new information to the analyzes of rational management of agricultural loess areas. I believe that the paper is worth publishing after making some minor adjustments, which I have listed below:
Special notes:
Markings: a and b in figure 2 must be in the picture, not the side Figure 3. I suggest introducing a scale (linear scale to each of the figures a, b, c) and linking them with arrows in a logical whole.
Figure 4. I don't really understand what the individual photos show. Can you explain ?
Figure 6 I think it is worth introducing line scales in satellite images.
Figure 8 why does the slope plot start with negative values? (-5)
Figure 9 The explanations partially fall into the choropleth areas. No linear scale. The callouts are sloppy. Have similar studies been made for loess areas in other parts of the world? There are no citations of literature from regions outside of China
Author Response
Dear editor
Thanks a lot for having reviewed our manuscript. Your comments were highly insightful and enabled us to greatly improve the quality of our manuscript. In the following pages are our point-by-point responses to each of the comments. Most of the revisions are in the manuscript. Some explanations regarding the revisions of our manuscript are as follows.
Response to Reviewer 2:
Thanks a lot for your affirmation of our research results and carefully comments on our paper. We have revised according to your comments:
1、a and b in figure 2 must be in the picture, not the side Figure 3. I suggest introducing a scale (linear scale to each of the figures a, b, c) and linking them with arrows in a logical whole.
Answer: Thank you very much for the careful suggestion. Figures 2 and 3 have been modified according to your suggestions. We have revised in the manuscript.
2、Figure 4. I don't really understand what the individual photos show. Can you explain?
Answer: We have paid attention to this issue. Figure 4 is the distribution of survey routes of villages in typical gully Basin. In order to reduce misunderstandings, we have deleted the picture.
3、Figure 6 I think it is worth introducing line scales in satellite images.
Answer: We quite agree with you that introducing line scales in satellite images, and revise the manuscript.
4、Figure 8 why does the slope plot start with negative values? (-5)
Answer: Thank you very much for the careful suggestion. Figure 8 shows that spatial variation of different types of gully farmland. The slope plot start with negative or positive values denotes that the distribution changes with distance.
5、Figure 9 The explanations partially fall into the choropleth areas. No linear scale. The callouts are sloppy. Have similar studies been made for loess areas in other parts of the world? There are no citations of literature from regions outside of China
Answer: We quite agree with you, and revise the figure 9. We sorted out similar studies been made for loess areas in other parts of the world and provide reference of literature from regions outside of China.
We will try our best to revise the paper!
Thanks a lot again for having reviewed our manuscript.
